

# Observations of Polar Mesospheric Summer Echoes Resembling Kilometer-Scale Varicose-Mode Flows

Jennifer Hartisch[1], Jorge L. Chau[1], Ralph Latteck[1], Toralf Renkwitz[1], and Marius Zecha[1]

[1]Leibniz Institute of Atmospheric Physics, Schlossstraße 6, 18225 Kühlungsborn, Germany

**Correspondence:** Hartisch (hartisch@iap-kborn.de)

**Abstract.** The mesosphere and lower thermosphere (MLT) region represents a captivating yet challenging field of research. Remote sensing techniques, such as radar, have proven invaluable for investigating this domain. The Middle Atmosphere Alomar Radar System (MAARSY), located in Northern Norway (69°N, 16°E), uses Polar Mesospheric Summer Echoes (PMSE) as tracers to study MLT dynamics across multiple scales. We recently discovered a spatiotemporally highly localized event showing a varicose mode, which is characterized by extreme vertical velocities ($|w| \geq 3\sigma$) of up to 60 m/s in the vertical drafts. Motivated by this finding, our objective is to identify and quantify similar extreme events or comparable varicose structures, i.e. defined by quasi-simultaneous up- and downdrafts that may have been previously overlooked or filtered. To achieve this, we conducted a thorough manual search through a MAARSY dataset, considering the PMSE months (i.e. May, June, July, August) spanning from 2015 to 2021. This search has revealed that these structures do indeed occur relatively frequently with an occurrence rate of up to 2.5% per month. Over the seven-year period, we observed and recorded more than 700 varicose-mode events and documented their vertical extent, vertical velocity characteristics, duration as well as their occurrence. Remarkably, these events manifest throughout the entire PMSE season with pronounced occurrence rates in June and July, while the probability of their occurrence decreases towards the beginning and end of the PMSE seasons. Furthermore, their diurnal variability aligns with that of PMSE. On average, the observed events persisted for 20 minutes, while the varicose mode caused an average expansion of the PMSE layer by a factor of 1.5, with a vertical expansion averaging around 8 km. Notably, a careful examination of the vertical velocities associated with these events confirmed that approximately 17% surpassed the $3\sigma$ threshold, highlighting their extreme nature.

## 1 Introduction

As the boundary layer between the Earth's atmosphere and outer space, the MLT region is in many aspects particularly interesting in terms of its dynamics and physical processes. On the one hand, the polar summer mesopause (78-88 km) is the coldest place in the Earth's system with temperatures as low as 130 K. These low temperatures are caused by dynamical processes such as the breaking of atmospheric gravity waves that bring the atmosphere out of thermodynamic equilibrium (Lübken et al., 1999). On the other hand, it is particularly difficult to continuously study this region of the atmosphere, as it is too low for satellites and too high for most in-situ measurement methods such as balloons. The use of remote sensing techniques, particularly radar systems, has proven as a notably advantageous and valuable approach in scientific MLT investigations. This



is primarily due to their capability to effectively and continuously probe the altitudes of interest while being independent of weather conditions and the time of the day.

In the polar summer mesopause, cold temperatures create ideal conditions for the formation, growth, and sedimentation of ice crystals of different sizes. Ice particles larger than $\approx 20$ nm can be observed with the naked eye as, e.g. the so-called noctilucent clouds (NLC), at an altitude of about 82 km. These clouds are visible to the observer at high latitudes as they are illuminated by the sun that has set during the night. Their occurrence and properties provide valuable insights into the composition and dynamics of the MLT region (Rapp and Lübken, 2004). Smaller ice particles are light enough to exist up to an altitude of around 90 km. They can be immersed in the D-region plasma, causing them to become charged. Additionally, in the altitude range of 80-90 km, gravity waves (GWs) are frequently observed propagating from lower atmospheric layers. These GWs become unstable, and generate turbulence in the mesosphere. The turbulent velocity field transports charged ice particles, leading to small-scale structures in the spatial distribution of both the charged particles and the attached electron number density. This transport process leads to the occurrence of irregularities in the radio refractive index, which is effectively determined by the electron number density at these altitudes. These irregularities can cause radar echoes at predominantly very high frequencies (VHF) observed from the ground, and are known as PMSE. Those are strongly influenced by the background conditions and thus by dynamical processes at different scales. Therefore, they are often used as tracers for dynamical processes within the MLT. A comprehensive overview of this phenomenon can be found in Rapp and Lübken (2004). However, it is important to emphasize that the occurrence of PMSE is subject to daily and seasonal variations (Latteck and Bremer, 2017), and that they are most frequently observed within an altitude range of 80 to 90 km. First echoes are typically seen around mid of May, while the occurrence maximizes in June and July and fades in August (e.g. Latteck et al., 2021).

PMSE serve as a unique tool for investigating the horizontal and vertical wind velocities in the MLT since the Doppler frequency shift of the radar echo is directly related to the vertical velocity component $w$ of the winds. By determining the frequency shift by using for example the complex autocorrelation function of the signal, the line-of-sight (radial) velocities can be determined providing valuable information about the vertical motion within the MLT region. Additionally, the spectral width reflects the distribution of velocities, allowing for the evaluation of turbulence (Woodman and Guillen, 1974). The former in particular is not easy to obtain (Gudadze et al., 2019), although it has a significant influence on the energy budget and the deposition of momentum in the atmosphere (e.g. Chau et al., 2021; Garcia and Solomon, 1985). Numerous studies suggest that much lower velocities can be expected in vertical winds compared to horizontal winds. For example, mean meridional and zonal wind speeds are of the order of several 10 m/s (e.g. Hoffmann et al., 2002; Jaen et al., 2022), whereas mean vertical wind speeds are typically below 5 m/s (e.g. Li et al., 2018; Chau et al., 2021). For this reason, most studies of mesospheric dynamics neglect vertical winds. Given this widespread assumption, the discovery of vertical wind velocities comparable to horizontal wind velocities was highly unexpected.

In 2021, Chau et al. (2021) reported a single event observed by MAARSY in July 2016 over northern Norway. This event was characterized by uncommonly high vertical velocities, limited spatially and temporally between 80 and 90 km altitude as shown in a corresponding Range-Time-Doppler-Intensity (RTDI) plot in Figure 1. Vertical velocities moving away from the radar are shown in red, blue represents vertical velocities moving towards the radar, and green represents vertical velocities near





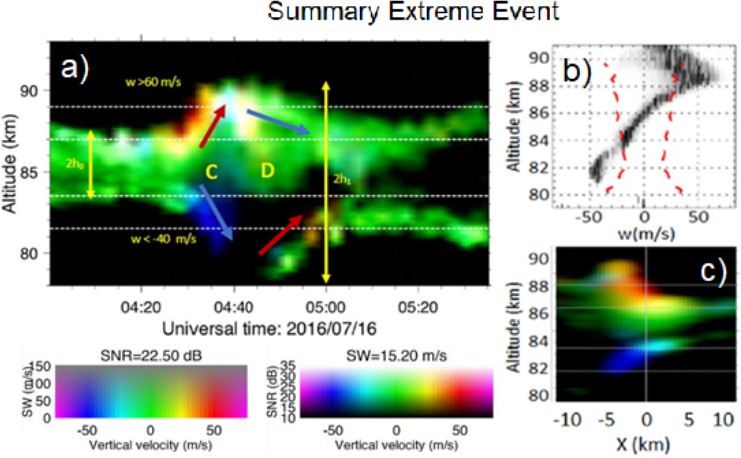

**Figure 1.** Summary of the highlights of an observed extreme vertical velocity event with MAARSY. (a) Time evolution of the vertical velocities between 75 and 95 km altitude over a time period of 1.5 hours. The red and blue colors indicate the strong up- and downdrafts, respectively, green indicates vertical velocities around 0 m/s, and the brightness of the colors represents the strength of the signal (SNR). (b) One example of the vertical velocity spectra during the event with the $3\sigma$ presented as the red dashed lines. (c) A cut through the event and showing the horizontal extent at one specific time (adapted from Chau et al. (2021)).

0 m/s. The intensity of the colors corresponds to the signal strength (SNR). The observed event, based on Doppler frequency measurements with MAARSY's vertical beam, shows simultaneous upward and downward movements, including vertical wind velocities reaching up to 60 m/s in the updraft (see Figure 1b). The extreme event visually resembles a mesospheric bore or soliton, with upper and lower surfaces oscillating out of phase by 180° as discussed by Dewan and Picard (1998), but with

exceptional vertical velocities and extents. The fact that bores can form in the mesosphere was reported by e.g. Dewan and Picard (1998); Fritts et al. (2020).

The discovery of the extreme event by Chau et al. (2021) provided motivation to explore older datasets of MAARSY to identify other extreme events. This motivation was further strengthened by the results presented in Feraco et al. (2018), where extreme vertical velocities under certain flow conditions were predicted based on direct numerical simulations (DNS). Ad-

ditionally, Dong et al. (2021) simulated vertical velocities of around ±40 m/s using the Complex Geometry Compressible Atmosphere Model for Polar Mesospheric Clouds (CGCAM-PMC). These recent findings have brought attention to the possibility that such high vertical velocities may have been overlooked or misinterpreted in previous analyses. Traditionally, outliers or anomalies in a dataset have been identified as individual values exceeding three times the standard deviation. These values have been traditionally treated as outliers (e.g. Chau et al., 2021).

Unpredictable short-term events with large impacts – in other words, extreme events – can occur in the upper atmosphere as well as in the lower atmosphere (e.g., tornadoes) and even in the ocean (e.g., tsunamis). While research exists for the latter, in-situ observation and studies of extreme events and small-scale dynamics in the polar MLT region are scarce.





Initially, the primary objective of this study was to investigate the frequency of events characterized by exceptionally high vertical velocities and small scales. It became apparent that such extreme events are not frequent. Instead, a recurring observa-

tion of events exhibiting a similar structure referred to as varicose (simultaneous upward and downward motion) emerged. The focus of this study is to provide a comprehensive report on these observations. As a new objective towards the ultimate goal, the aim is to gather as many characteristics as possible related to the occurrence of these events, particularly focusing on the associated vertical velocities, which originally motivated this work.

The data set and methods used in this study are described in Sect. 2. The findings resulting from this analysis are presented

in Sect. 3 and discussed in Sect.4.

## 2    Database and Method

This study uses PMSE measurements obtained from MAARSY, a mesosphere/stratosphere/troposphere (MST) radar located on Andøya in northern Norway (69.30°N, 16.04°E). MAARSY is a monostatic, active phased array radar operating at a frequency of 53.5 MHz. Its design allows high flexibility in beam forming and beam steering as well as the use of modern

interferometric applications for improved studies of the Arctic atmosphere with high spatiotemporal resolution (Latteck et al.,

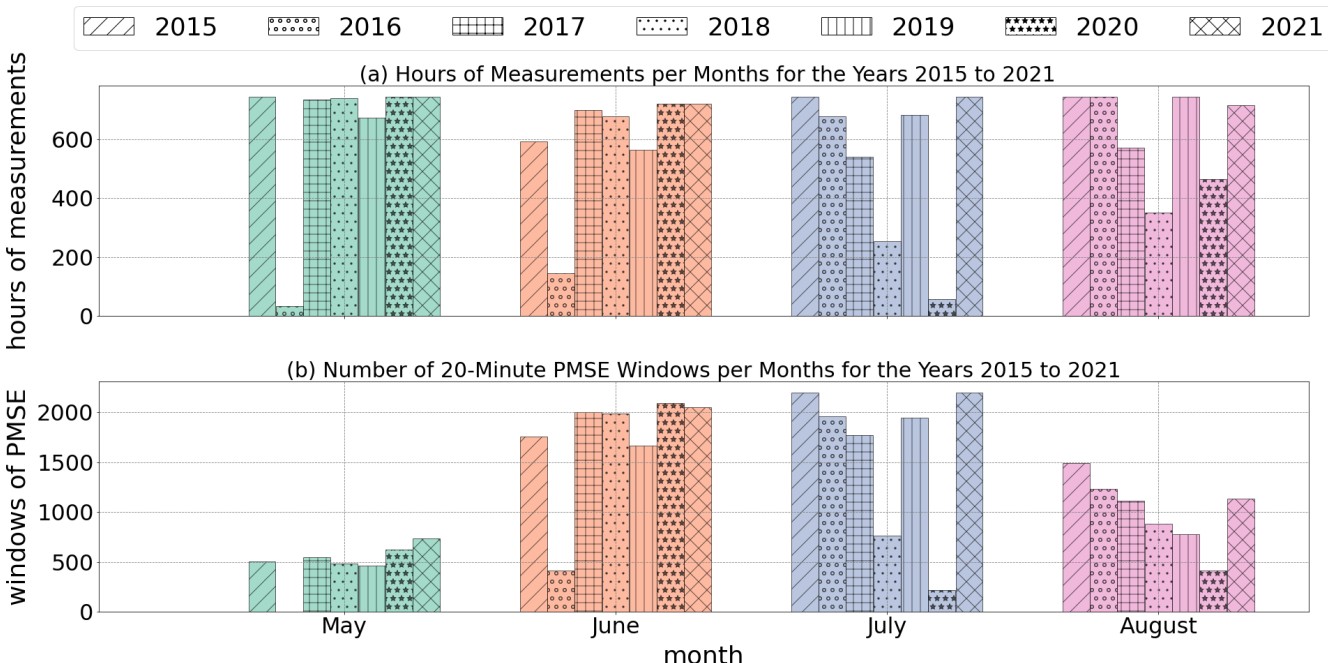

**Figure 2.** (a) Number of hours during which the radar was operational and included in the data analysis and (b) Number of 20-minute windows in which PMSE were detected and included in the data analysis, separately for each month of 2015 to 2021.



2012). A collection of the main technical parameters of MAARSY and of the experimental configuration as used for the standard observation of PMSE is given in Latteck and Bremer (2017).

The dataset used in this study includes PMSE observations during the PMSE seasons (i.e. May, June, July, and August) of 2015–2021 with 766 observation days. Notably, the number of observation hours was not constant, as shown in Figure 2a. Note that the measurement hours cover the radar's entire operational duration, including periods without PMSE detection (see Figure 2b). The partly limited number of observation hours needs to be taken into consideration for the later interpretations of the results. In the years 2017, 2019, and 2021, nearly every day was included in the analysis. However, the first half of the PMSE season in 2016 and the second half of the seasons in 2018 and 2020 included only about one-third of the days. During these periods, special experiments were carried out with MAARSY characterized by a very low Nyquist frequency, which does not allow an analysis to be carried out with regard to extreme events.

The data used in this study are based on different radar experiments, all involving the vertical beam-pointing direction. A pulse repetition frequency (PRF) of 1 kHz and two coherent integrations, resulting in a Nyquist frequency of $f_N = 250$ Hz, were predominantly used, enabling the determination of maximum vertical velocities of up to 700 m/s. Exceptions from that are three months (June-August) in 2018, one month (July) in 2019, and one month (July) in 2020 where the PRF was adjusted to 828 Hz, 850 Hz and 900 Hz, respectively, which already modifies the .

In previous radar experiments, the raw data were additionally subjected to a high number of coherent integration before undergoing further signal processing. This integration aimed to manage the amount of data effectively (Farley, 1985). However, this approach reduces $f_N$ and can lead to the filtering of unexpectedly high vertical velocities, as shown in Figure 3. Since 2017, the standard MAARSY experiments for observing PMSE have avoided additional coherent integrations, allowing for the reanalysis of a significant portion of the MAARSY dataset using the full Nyquist frequency $f_N=250$ Hz which corresponds to a maximum detectable radial velocity $v_{rad} = 700$ m/s. However, the raw data from years prior to 2017 had a reduced Nyquist frequency, with values of $f_N = 15.64$ Hz (equivalent to $v_{rad} = 43.8$ m/s) in 2015 and 62.5 Hz (equivalent to $v_{rad} = 175$ m/s) in 2016 which allowed the discovery of the extreme event presented by Chau (2021). Data before 2015 were excluded from this study due to the substantial reduction in $f_N$ caused by a high number of coherent integrations.

The search for additional extreme events has presented notable challenges, which arises from the stochastic nature of meteor occurrences, characterized by their unpredictable and sudden appearance with remarkably high radial velocities in the measurements. Additionally, system-induced artifacts from higher range gates during instances of exceptionally intense PMSE further complicate an automation process. As a result, the identification of conspicuously high vertical wind velocities required manual inspection.

Following model expectations (e.g. Feraco et al., 2018; Dong et al., 2021), events similar to the extreme event reported in Chau et al. (2021) were found to be infrequent. Interestingly, lower-velocity events with the same varicose-mode structure, however, occurred more frequently. Varicose-mode events were considered to be those phenomena that had simultaneous vertically ascending and descending winds and lasted only a few minutes. To clarify the manual event selection process, Figure 4 presents selected events with simple schematics. The left column shows the example events in the form of aforementioned RTDI plots. The varicose structure is clearly evident in all three instances. Similar to the extreme event in Figure 1, this type





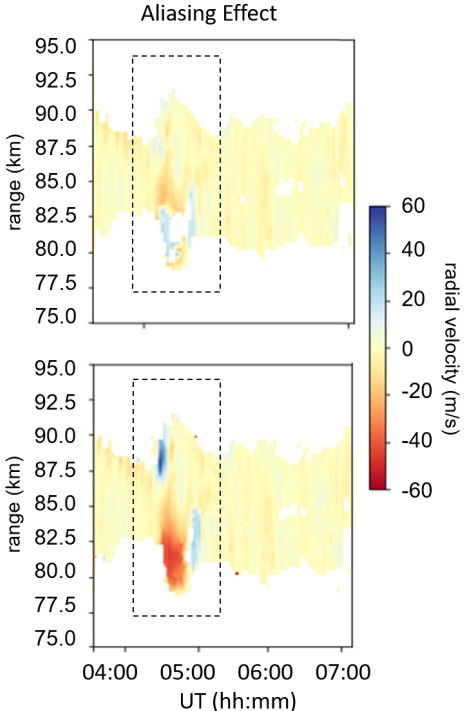

**Figure 3.** The colored contours show the vertical velocity (a) after applying 6 additional coherent integrations (effective $v_{rad}$ = 140 m/s) and (b) without applying additional coherent integrations (effective $v_{rad}$ = 175 m/s) over altitude. The velocities cannot be determined from the spectra due to filtering and thus are missing in (b).

of structure causes the PMSE layer to expand during its presence and contract afterward. The outer boundaries of the PMSE layer are indicated by a black line, which is colored red and blue for up- and downdrafts, respectively in the right column of Figure 4. The vertical extent of the PMSE layer before and during the events is represented by $2h_0$ and $2h_1$, respectively. The start time of each event is marked as $t_0$, and the duration of an event is denoted as d.

To address challenges in varicose-mode event observations related to diurnal and annual PMSE variability, and fluctuating observation hours, the event numbers are normalized to the occurrence of PMSE within a specific time period. PMSE within the MAARSY dataset was classified as such if it met specific criteria. The normalization process involved dividing the resulting daily PMSE-defining matrices into twenty-minute windows, focusing on the altitude range of 82 to 88 km with high PMSE occurrences. These restricted windows were analyzed for PMSE, considering a 50% threshold of PMSE-labeled pixels across

all altitudes to indicate PMSE presence.





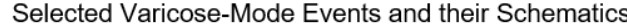

**Figure 4.** Selected varicose-mode events: Vertical velocity evolution (color-coded: red means upwards, blue means downwards, green means |w| around 0 m/s) between 75 and 95 km, exhibiting simultaneous up and downwards motions. Color brightness indicates signal strength. The observations were made on (a) June 5, 2020; (b) June 13, 2021; and (c) June 26, 2021. The right column shows event schematics with $h_0$ and $h_1$ representing undisturbed and disturbed PMSE layer thickness, respectively. $t_0$ denotes event start time, and d represents duration. Red and blue colors indicate vertical wind direction, as in the left column



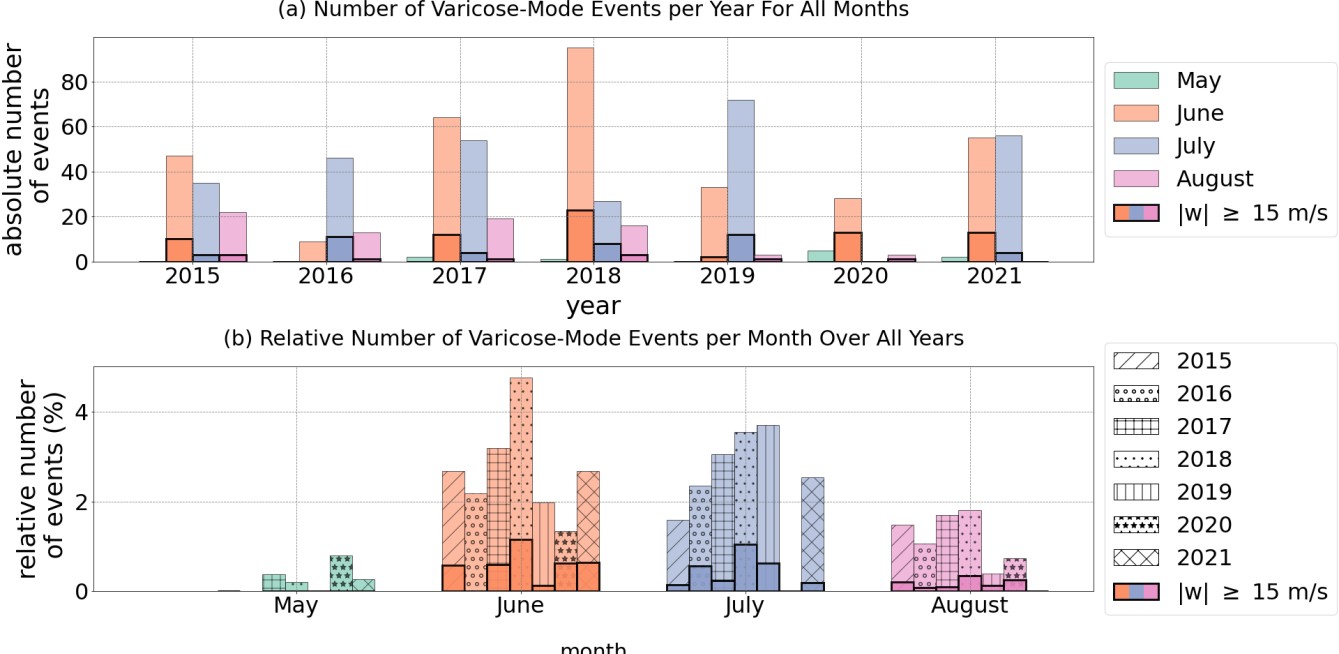

**Figure 5.** Occurrence of varicose-mode events in the PMSE months (May, June, July, August) between 2015 and 2021. (a) The absolute number of events counted per year, broken down by individual months. (b) The number of events counted per month relative to the number of 20-minute PMSE windows counted during the same period, broken down by individual years. The more heavily colored and contoured bars represent the absolute (relative) number of those events in the respective period with |w| ≥ 15 m/s.

## 3 Results

The analysis of the MAARSY dataset (2015-2021) revealed frequent occurrences of varicose-mode events. Figure 4 illustrates three characteristic events to demonstrate the recurring properties of typical varicose structures and their duration. These events are not limited to a single occurrence within a short period of time (e.g., 1 hour), as shown in Figure 4a, where structures can

occur successively with only a few minutes to a few tens of minutes in between. The plots also highlight that while the structures themselves appear similar, their vertical extent and the presence of substructures, such as additional layers, vary from case to case. Information regarding occurrence, duration, vertical extent of the PMSE layer, and prevailing vertical wind speeds was collected for the identified events.

### 3.1 Varicose-Mode Event Occurrence

The analysis of the 766 observation days showed that varicose structures occurred on 249 days, corresponding to an overall occurrence rate of about 33%. Within the seven PMSE seasons, 707 varicose-mode events were observed. A detailed overview





of the occurrence of these events during the observation period is given in Figures 5a and b and Figure 6a. Supplementary data summarizing the values can be found in Table 1.

**Table 1.** Summary of the absolute numbers of varicose-mode events per month and year. In parentheses the number of 20-minute windows that contained PMSE for the same time period. In the last column and row the total numbers are shown, respectively.

| month / year | 2015 | 2016 | 2017 | 2018 | 2019 | 2020 | 2021 | total per month |
|:---:|:---:|:---:|:---:|:---:|:---:|:---:|:---:|:---:|
| **May** | 0 (504) | 0 (0) | 2 (543) | 1 (484) | 0 (460) | 5 (626) | 2(738) | 10 (3355) |
| **June** | 47 (1755) | 9 (412) | 64 (2005) | 95 (1992) | 33 (1664) | 28 (2097) | 55 (2054) | 331 (11979) |
| **July** | 35 (2202) | 46 (1961) | 54 (1771) | 27 (762) | 72 (1947) | 0 (216) | 56 (2201) | 290 (11060) |
| **August** | 22 (1489) | 13 (1233) | 19 (1115) | 16 (885) | 3 (774) | 3 (414) | 0 (1133) | 76 (7043) |
| **total per year** | 104 (5950) | 68 (3606) | 139 (5434) | 139 (4123) | 108 (4845) | 36 (3353) | 113 (6126) | **707 (33437)** |

Varicose events can occur as single events, pairs (two events), or groups (more than two events), as shown in Figure 4, and the absolute numbers of them being categorized as single events, pairs, and groups, is shown in Figure 6b for each year. The bars with more intense colors and thicker black frames represent the total number of events during each period with $|w| \geq$ 15 m/s. In the case of Figure 6b, this number is increased by one count if a single event within a pair or group has such high vertical velocities. However, the presence of multiple events greater than 15 m/s within a pair or group is not counted separately. By far the majority of events in the varicose mode occurred as single events, and more than two events within a shorter period were observed least frequently.

Figure 5a shows the absolute number of events per year, broken down by each month of the PMSE season. Events with vertical velocities $|w| \geq$ 15 m/s are visually highlighted by more intense colors and a thicker black frame. The value of $|w| \geq$ 15 m/s has been selected as a conservative number that represents at least $3\sigma$ of standard values. It is noteworthy that events in varicose mode were observed in all years, with no discernible pattern. However, the total number of events per year showed variation, with a relatively high number of events observed in 2018 and only a few in 2020. In addition, the proportion of each PMSE month to the total number of events varied across the years.

Figure 5b shows the relative frequency of varicose-mode events versus the number of 20-minute PMSE windows to provide a normalized perspective. Each month is shown with a unique color, while different patterns represent individual years. Most PMSE events occurred in June and July, with fewer events in August and the lowest frequency in May. On average, the probability of observing a varicose-mode event during the PMSE season was approximately 2.5% in June and July, decreasing to 0.3% at the beginning and increasing to 1.0% towards the end of the season, given that PMSE were detected.

The diurnal variation of varicose-mode events is shown in Figure 6a as absolute numbers spread over a day throughout the 7-year period. The graph illustrates that these events can occur at any time of day. However, a clear peak is observed between 12:00 and 16:00 UT, while a minimum occurs between 18:00 and 20:00 UT. To highlight the number of events with $|w| \geq$ 15 m/s, bars with a more intense color and a thicker frame are used in Figure 6a.



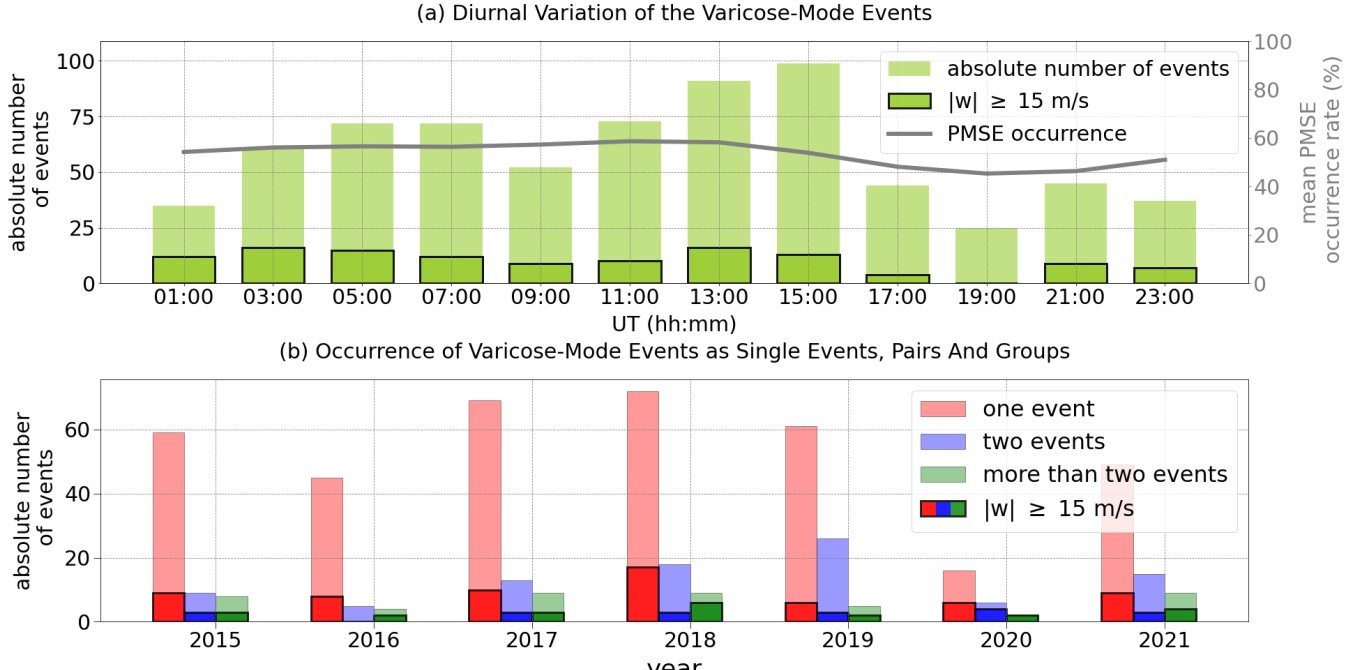

**Figure 6.** (a) The distribution of the occurrence of events in varicose mode over the course of a day. The gray line illustrates the mean diurnal variation in the frequency of PMSE by Latteck et al. (2021). (b) The absolute number of varicose mode events occurring as single events, as pairs, or in groups for each individual year. The more heavily colored bars represent the relative or absolute numbers in (a) or (b) of events with $|w| \geq 15\,\text{m/s}$.

## 3.2 Varicose-Mode Duration and Vertical Extent Characteristics

Figure 7a shows the durations (see d in Figure 4) of the events found in this study. It was discovered that the events typically lasted for an average of $(20 \pm 9)$ minutes (see red solid line in Figure 7a), ranging from a minimum of 4 minutes to a maximum exceeding 100 minutes. The aforementioned PMSE layer characteristics during the events are summarized in Figure 7b and

c, showing the vertical expansion variations caused by the varicose structures. The red solid line in Figure 7b represents the average maximum vertical PMSE layer thickness (see $2h_1$ in Figure 4), which is approximately $(8 \pm 2)\,\text{km}$, with values ranging from 3 to 15 km. The ratio between the initial vertical width of the PMSE layer (see $2h_0$ in Figure 4) and its maximum width during an event ($2h_1$) is depicted in Figure 7c indicating a mean vertical widening by a factor of about $1.5 \pm 0.5$, with a range spanning from a minimum of 0.3 to a maximum of 7. The respective occurrence and mean values for events with $|w| \geq 15\,\text{m/s}$,

are represented by the more intensely colored bars and the red dashed lines in Figure 7b and c.



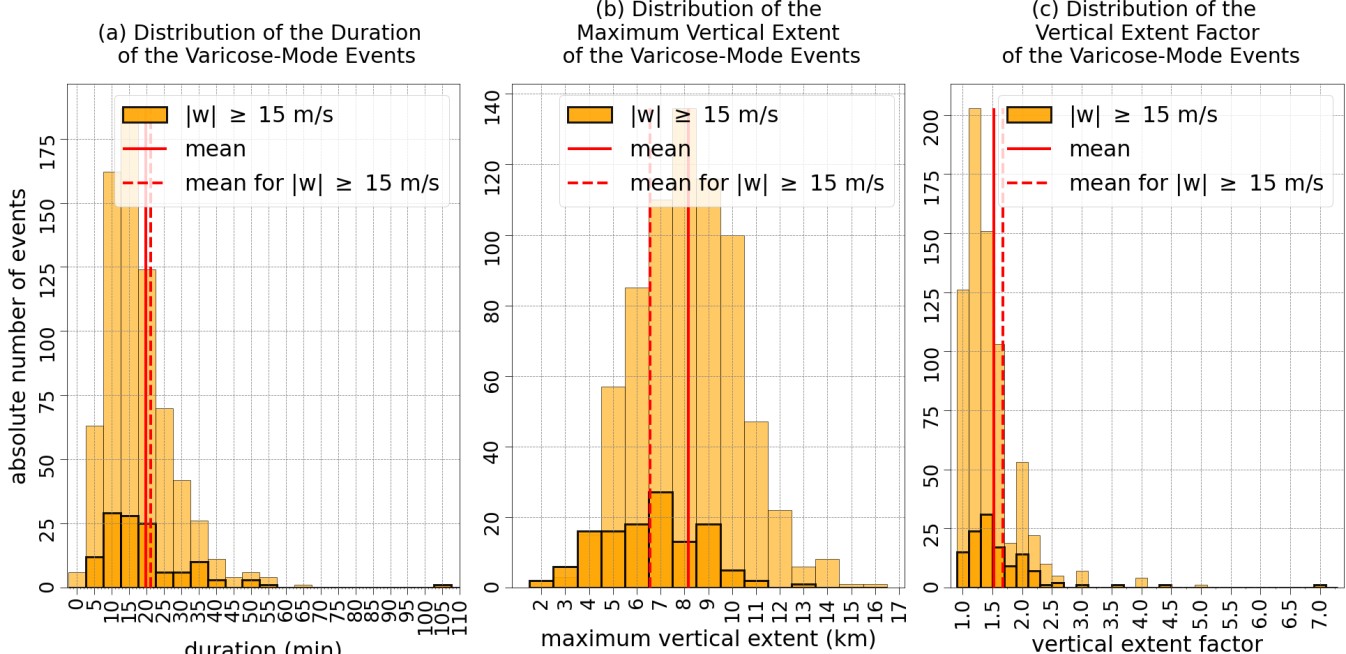

**Figure 7.** Duration (a), maximum vertical extent (b) and $h_1/h_0$ (c) distribution of the varicose-mode events. The more heavily colored bars represent the absolute numbers of events in the respective period with $|w| \geq 15$ m/s. The red solid (dashed) line represents the mean of all events ($|w| \geq 15$ m/s events).

### 3.3 Vertical Velocities

The main focus of this study was to gain a comprehensive understanding of the variability of vertical wind velocities. In order to accurately interpret the velocities of individual events, it is important to first examine the overall distribution of velocities over the entire period and height range (82 to 88 km for 7 PMSE seasons). This velocity distribution, shown in green in Figure 8a, follows a normal distribution with the center at 0 m/s and a standard deviation of about 3 m/s. However, closer inspection reveals that the distribution is not completely symmetric and that downward vertical velocities (negative values) predominate.

Figure 8b shows the velocity distributions for the highest upward and downward velocities recorded during each varicose-mode event, represented by red and blue bars, respectively. It is clear that most of the vertical velocities of the varicose-mode events fall within the range of about $\pm$ 15 m/s. The black line compares these values to a normalized Gaussian reference, mirroring the distribution in Figure 8a.

The vertical velocity distribution of varicose-mode events shows a wide range, with updrafts exhibiting a slightly broader distribution compared to downdrafts. In total, velocities of $|w| \geq 15$ m/s were observed in 67 downdrafts and 90 updrafts. In 34 cases these velocities were observed in simultaneously in the up- and downdraft.





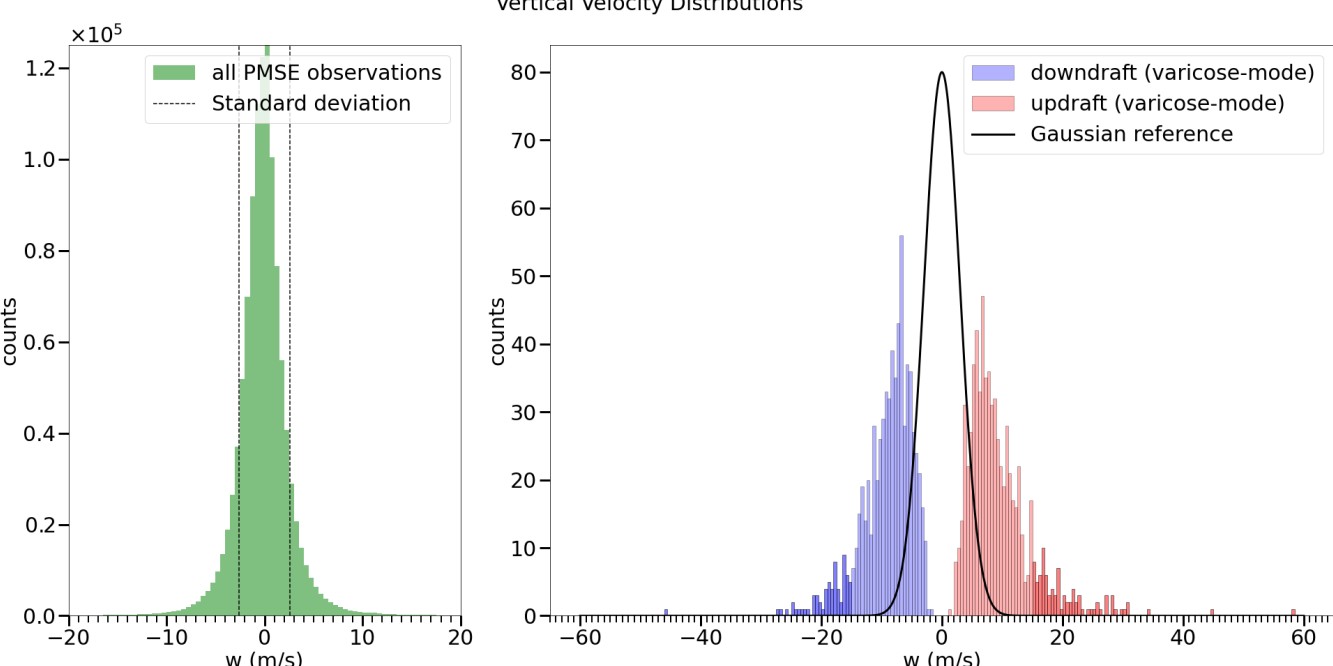

**Figure 8.** Distribution (a) of vertical velocities across all altitudes and throughout the observation period and (b) of the maximum vertical velocities of varicose-mode events in blue (down-drafts) and red (up-drafts). The bars with more intense colors indicate the absolute number of events with |w| ≥ 15 m/s. The Gaussian curve in (b) represents the normalized distribution shown in (a).

### 3.4 High-velocity varicose-mode events

The distribution of vertical velocities in varicose-mode events is broad, with high-velocity events (|w| ≥ 15 m/s) accounting for about 17% of the total varicose-mode event collection. Previous figures showing the distribution of all varicose-mode events over different time scales (year, month, or day), as well as providing information on their duration, vertical extent, and vertical extent change, also show events characterized by vertical velocities of |w| ≥ 15 m/s. These are highlighted in the figures by more intense colors and thicker outlines of the bars.

It was found, that high-velocity varicose-mode events also occurred every year and can be observed in any month of a PMSE season except May (see Figure 5a and b) and at any time of the day except for 18:00 UT (see Figure 6a). Additionally, they can be found as single events, in pairs, or in event groups (see Figure 6b). On average, these remarkable events last about (22 ± 13) minutes (see Figure 7a), reach a maximum vertical extent of about (6.5 ± 2) km (see Figure 7b), and experience a change in vertical extent by a factor of about 1.7 ± 0.7 (see Figure 7c).

Figure 8b shows that updrafts have more absolute velocities above 15 m/s than downdrafts (42 vs. 39). While these numbers seem very similar, the distribution of events with |w| ≥ 15 m/s appears to be broader for upward winds, because of a higher number of velocities above 20 m/s. Specifically, there are 7 downwind and 17 upwind events in the latter category.



## 4 Discussion and Outlook

Motivated by the occurrence of an extreme event showing high vertical velocities in a varicose structure, this study has un-
210 dertaken an extensive manual search for other events. It has become evident that the predictions made by DNS and models
held true and that such events do not occurr frequent. The study encompasses a dataset spanning the PMSE months between
2015 and 2021. However, the significance of this study lies in the observation of localized, temporally restricted structures
characterized by a varicose mode.

### 4.1 Varicose-Mode Events and Instabilities

In comparison to other studies, these structures bear the closest resemblance to mesospheric bores, a concept proposed by
Dewan and Picard (1998). Their study introduced the term internal mesospheric undular bore to describe an event observed
in airglow measurements, characterized by a sharp wave crest followed by smaller trailing waves. This event had contrasting
effects on different layers of the atmosphere. It increased the brightness of the OH layer at 85 km and the Na layer at 90 km,
but had the opposite effect on the O and OI layers at 94 km and 96 km, as reported by Taylor et al. (1995). Dewan and Picard
(1998) drew parallels between these airglow observations and phenomena observed in the troposphere and water, such as
undular tidal bores in river beds and "morning-glory" clouds. They suggested that these mesospheric bores require a channel-
like structure, potentially provided by a thermal or Doppler duct, caused by a temperature inversion layer (also known as
mesospheric inversion layer, MIL) or wind shear. The theory of a duct for a guiding structure was initially confirmed by
She et al. (2004) through the observation of temperature inversion layers coinciding with the occurrence of two mesospheric
undular bore events over Fort Collins, Colorado in 2002. This finding is further supported by Hozumi et al. (2019) who studied
space-borne airglow observations and found that most mesospheric bores observed between 55°N and 55°S occurred within
such inversion layers. In contrast to the observations made by Dewan and Picard (1998), the trailing waves as the bore-defining
feature is not evident in all of the observations in this study. Instead, they mostly appear as unattached wave crests, characteristic
of solitary waves that occasionally result from bores (Koch et al., 2008).

In this study, the observed events can be described as solitary waves in a varicose mode, meaning quasi-simultaneous
upward and downward movements (Lighthill, 1979). To investigate potential ducting mechanisms during the observed events,
temperature measurements within a small radius around the MAARSY site and close-range measurements of the horizontal
wind components are essential. Fortunately, data from the Spread-Spectrum Interferometric Multistatic Meteor radar Observing
Network (SIMONe), operated by the Leibniz Institute of Atmospheric Physics (IAP) in close proximity to MAARSY, can be
utilized for this purpose. Additionally, data from an Iron Lidar operated by IAP, located near MAARSY, could be examined to
identify mesospheric temperature inversion layers in the future.

A further step towards understanding the underlying physics of these varicose (including extreme) events is to use the
collected background conditions to attempt a replication of the varicose-mode structures and the exceptionally high vertical
velocities through DNS. Initial attempts have been recently performed by Ramachandran et al. (2023). Using a two-dimensional
DNS of the Navier-Stokes equation with the Boussinesq approximation, they successfully replicated key characteristics of



a mesospheric bore event observed in OH airglow over Kühlungsborn, Germany in March 2021. Their findings highlight the influential role of the temperature duct width and the amplitude of the initial perturbation in shaping the evolution and morphology of the bore.

## 4.2 Occurrence Statistics

The analysis of more than 700 varicose structures in the MLT region revealed an overall occurrence rate of approximately 33%. Although there is no discernible pattern in events across years, fluctuations in the total number of events per year were observed. The higher occurrence of events in 2018 and the relatively few events in 2020 demonstrate this variability. This lower event counts in 2020 (see Figure 5a) may be attributed to a reduced overall observation time (see Figure 2a). But this explanation is challenged by the maximum number of events observed in 2018, despite a comparatively low number of observation hours.

The implications of this discrepancy require further investigation. Individual months also contributed differently to the total occurrence, highlighting the influence of temporal factors on the occurrence of varicose events.

The concentration of events in the middle of the PMSE season, particularly in June and July, is consistent with previous studies that have identified these months as peak periods for PMSE activity. It is important to note that varicose-mode event occurrence was analyzed in relation to the number of 20-minute PMSE windows, and a normalization factor was applied to

255 account for diurnal and seasonal variations in PMSE occurrence. This normalization ensures the validity and comparability of event counts per PMSE season or month. The methodology included validation of PMSE occurrence within specific time periods using a threshold-based approach (see Section 2). The density of events is thus higher in the second half of the PMSE season than in the first half. When considering events as individual occurrences, pairs, or groups, the relative numbers of high-velocity events increase, although their absolute numbers are lower.

The observed diurnal pattern, with a maximum occurrence between midday and 16:00 UT and a local minimum between 18:00 and 20:00 UT, aligns with the known diurnal variability of PMSE over Andøya (e.g. Hoffmann et al., 1999; Latteck et al., 2021). The significant variability in event counts relative to the average rate of PMSE occurrence strongly suggests a compelling link between varicose-mode events and the factors influencing PMSE patterns. This implies that both phenomena likely share common underlying mechanisms, indicating a potential interdependence between them. However, the limited

dataset and reliance on PMSE as a tracer prevent conclusive statements about annual, inter-annual, or diurnal periodicity.

If assuming these events are mesospheric bores, comparisons can be drawn regarding occurrence rates to a recently published study by Hozumi et al. (2019), who examined the occurrence probabilities of mesospheric bores observed using airglow imagers on the International Space Station. Note, that the absence of evidence confirming the varicose structures as bores and the limited satellite observations (55°S to 55°N) outside MAARSY's range should be considered. Nevertheless, this suggests

that if the structures found in this study are bores, they may also occur frequently in mid-latitudes and equatorial regions, with a preference for mid- and equatorial latitudes during equinox seasons. Regarding diurnal variations, Hozumi et al. (2019) found a stronger dependence on daytime at mid-latitudes compared to near the equator.

The study, initially motivated by a singular extraordinary event, unexpectedly discovered occurrences of events in pairs or groups. The close time intervals between individual events within a pair or group suggest conditioning and influence from one





event to the next. The possibility of memory involvement and differences in the origin of isolated events versus events in pairs or groups are intriguing. Group events exhibit a distinct pattern with a leading wave crest and trailing waves, resembling bores. However, the amplitudes of the supposed trailing waves are nearly equivalent to the leading wave (see Figure 4a), contrary to previous reports (e.g. Taylor et al., 1995; Dewan and Picard, 1998). The significance of these statistics is yet to be determined.

The occurrence patterns of high-velocity events ($|w| \geq 15$ m/s) and varicose-mode events in general align, but there is no
direct correlation with the absolute (Figure 5a) or relative (Figure 5b) number of the latter. In other words, a higher number of varicose-mode events does not necessarily correspond to a higher number of high-velocity events. An example can be seen in Figure 5a for 2021, where the absolute number is higher in July compared to June, but this behavior is not mirrored by the events with absolute vertical velocities exceeding 15 m/s – there are significantly more of those in June than in July. Similarly, in the year 2020, no high-velocity events were observed in May, despite a larger absolute number of events compared to Au-
gust. This lack of correlation is further highlighted in Figure 5b, where it becomes even more apparent that years with a higher number of varicose-mode events per number of PMSE windows, such as in July, do not necessarily exhibit a higher occurrence of events with higher vertical velocities. The diurnal variation of high-velocity events is less pronounced compared to general varicose events, with a notable absence of them between 18:00 and 20:00 UT (see Figure 6a).

### 4.2.1 Duration

The duration of varicose-mode events, averaging 20 minutes, demonstrates their relatively short-lived nature. This finding justifies the selection of a 20-minute window size for counting PMSE occurrences, as it captures the majority of these events. Compared to the event described by Chau et al. (2021), which lasted approximately 25-30 minutes, this average duration is a bit smaller.

It is important to note that the radar illumination area in the mesosphere is relatively narrow (5.4 km diameter at 85 km altitude), and dynamical structures only pass through it, making them visible for a duration that depends on their speed through the field of view. The observed durations and vertical expansion variations in varicose-mode events provide valuable insights into their characteristics and potential underlying mechanisms. Mesospheric bores, for instance, can persist for several hours (e.g. simultaneous airglow and lidar observations by Smith et al., 2017; Fritts et al., 2020). The small observation volume in
the mesosphere makes it challenging to determine exact lifetimes and to draw conclusions about the production and dissipation of the varicose-mode events.

In terms of duration, high-velocity events exhibit a similar average duration of around ($22 \pm 13$) minutes compared to general varicose events, but with a larger standard deviation. This prompts further investigation into the potential correlation between maximum vertical velocity and event duration.





### 4.3 Vertical and Velocity Characteristics

#### 4.3.1 Vertical Extent and Extent Change

The maximum vertical extent of the PMSE layer in varicose-mode events of $(8 \pm 2)$ km is smaller compared to the 12-km extreme event reported by Chau et al. (2021). Similarly, the expansion factor of the PMSE layer due to the varicose structure is with a mean value of 1.5 significantly smaller than the factor of 3 observed in the extreme event. Notably, there is also a difference observed in the extent factor between high velocity-events and general varicose-mode events. The former is slightly smaller $(1.7 \pm 0.7)$ than the latter. The contrary effect was found for the maximum vertical extent of the PMSE layer. The value for events with $|w| \geq 15$ m/s lies at approximately $(6.5 \pm 2)$ km, which is about 20% less than the value for all events. The mechanisms influencing the expansion factor in varicose-mode events are not fully understood. It is believed that vertical velocities, horizontal wind properties, and temperature profiles play a role in determining the propagation duct in which these events are most likely to travel (e.g. Ramachandran et al., 2023).

The majority of bore observations are conducted using airglow, lidar, or satellites, providing a two-dimensional view in space. In such cases, only the initial height $h_0$ is known by using, for example, half of the width of a temperature inversion layer (e.g. She et al., 2004), while the disturbed height $h_1$ (for $h_0$ and $h_1$ see Figure 4) remains unknown. To calculate the amplitude of a bore crest, a normalized bore amplitude (also referred to as bore strength) $\beta$ of 0.3 is normally assumed based on models. This value serves as a measure to differentiate between turbulent and undular bores and is determined by using $\beta = (h_1 - h_0)/h_0$ (e.g. Lighthill, 1979; Dewan and Picard, 1998). Radar-based bore observations allow for the direct determination of $2h_1$, avoiding assumptions in our observations. An average $\beta$ of 0.53 for all observed varicose-mode events can be seen in Figure 9 (see the red solid line). Significantly, events with $|w| \geq 15$ m/s show a slightly higher average $\beta$ of 0.68. This implies that if these structures are indeed bores, they exhibit a turbulent nature and the potential for wave breaking. However, an alternative theory proposed by She et al. (2004) suggests that for internal bores, a $\beta > 0.3$ can still sustain an undular bore configuration, as energy may leak out of the duct region.

Radar observations are limited to the radar field of view, making it uncertain whether these observed structures dissipate outside the observed area. A comparison to findings by Fritts et al. (2020) shows a similar $\beta$ (0.7) in their study of bore observations done simultaneously by an all-sky imager and lidar. Although such a value suggests a turbulent solitary wave, the observation revealed undular bores, possibly due to the absence of a thermal duct supporting a soliton development. This highlights the importance of investigating ducting conditions in case studies of the events analyzed in this study.

#### 4.3.2 Vertical Velocities

Improving the accuracy of vertical velocity measurements using radar can be challenging due to several factors that contribute to uncertainty. The radar system itself contributes to some degree due to systematic errors, such as beam width and beam orientation. While it is possible to determine vertical velocities by analyzing the Doppler shift of the received signal when the scatterer is directly over the vertically aligned radar beam, it is important to consider the limitations. Even with MAARSY's relatively narrow beam width, which is about $3.6°$ ($\approx 5.4$ km at 85 km), there is a possibility that scatterers will be detected





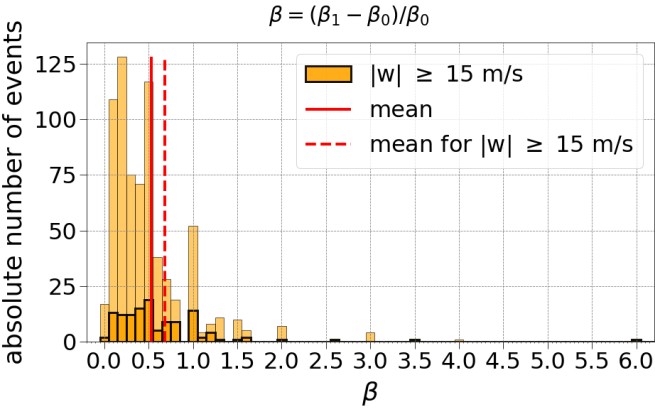

**Figure 9.** Normalized bore amplitudes of the varicose mode events in general. The more heavily colored bars represent the absolute numbers of events in the respective period with $|w| \geq 15$ m/s. The red solid (dashed) line represents the mean of all events (events with $|w| \geq 15$ m/s)

that are not perfectly over the center of the radar. Therefore, there is always some probability that the measured radial wind will contain a horizontal wind component. To minimize this effect, MAARSY's beam width is calibrated well and as narrow

as possible according to Renkwitz et al. (2017). It can thus be assumed, that the velocities measured consist mostly of the vertical component of the radial velocity, particularly for velocities greater than a few m/s. The measured vertical velocities in this study can be seen as the uppermost estimate of the vertical wind. Additionally, the identification of events in this study relied on visual assessment, which has inherent limitations. The choice of the color bar used in the visualization determined the detectability of varicose-mode events. A color bar ranging from -15 to 15 m/s was selected to focus on events exceeding

the $3\sigma$ threshold, but it may have constrained the identification of events with lower velocities, which was deemed unnecessary given the study's objectives.

The observed mean vertical velocity of the varicose-mode events of 10 m/s is already twice as high as the expected maximum vertical velocity in the MLT, which was anticipated to lie within w = ± 5 m/s. In light of the extreme event reported by Chau et al. (2021) and the model predictions mentioned earlier, it is not surprising to observe instances with $|w| \geq 15$ m/s. These

magnitudes are notably high, surpassing more than six times the standard deviation of $\approx 2.5$ m/s (see black dashed lines in Figure 8). It is important to note that in the study conducted by Chau et al. (2021), vertical velocities were classified as "extreme" when exceeding five times the standard deviation. One could thus refer to the high-velocity events of this study as "extreme" as well. Furthermore, it is observed that the occurrence of winds surpassing a threshold of ± 15 m/s is comparable between updrafts and downdrafts. Nevertheless, downdrafts exhibit a narrower range of velocities compared to updrafts, primarily due to

a larger quantity of updrafts with velocities exceeding 20 m/s. This could be explained by the fact that in the case of downdrafts, regions with higher temperatures are quickly reached, leading to the melting of ice particles and subsequent disappearance of PMSE. As a result, the tracer necessary to determine vertical velocities is absent in such cases.

The general asymmetry of vertical velocities observed in this study (more downwards than upwards directed velocities in Figure 8a) is consistent with previous findings. Gravitational sedimentation of ice particles forming PMSE, as described by





Gudadze et al. (2019), contributes to this asymmetry. The use of PMSE as tracers in the vertical radar beam can introduce a bias towards downward-directed vertical velocities, as discussed by Hoppe and Fritts (1995) who used the EISCAT VHF radar. The authors suggest that gravity waves can cause a tendency to underestimate the upward motion when using VHF radars. This is because there is a negative relationship between the vertical motion caused by gravity waves and the radar's ability to detect them (radar reflectivity). They found that there is a correlation between certain types of vertical motions and

radar measurements at different frequencies. For example, turbulence generated by upward-moving gravity waves can disrupt the radar's ability to measure them accurately. Additionally, it is plausible that in the case of downdrafts, regions with higher temperatures are quickly reached, leading to the melting of ice particles and subsequent dissolution of PMSE. As a result, the tracer necessary to determine vertical velocities is absent in such cases. Therefore, it is possible that the observed asymmetries may not be a true representation but rather a consequence of these influencing factors (e.g. Hoppe and Fritts, 1995; Gudadze

et al., 2019).

To enhance the understanding of varicose-mode occurrence statistics, further analysis of additional MAARSY data, including previously excluded data, and data from preceding radar systems such as the ALomar WINd radar (ALWIN), is necessary. Expanding the research scope to incorporate other high-latitude MST radar data sources would also provide valuable insights. Considering the events' small-scale structure, radar imaging offers detailed insights into their morphology and spatial pa-

rameters in the horizontal plane, along with their motion direction within the radar beam. This imaging technique enables the resolution of smaller structures within a larger observation volume (e.g. Sommer and Chau, 2016). By using imaging combined with the coherent MIMO ( Multiple Input Multiple Output) the angular resolution can be enhanced even further to around 1 km (Urco et al., 2019). Chau et al. (2018) presented another way of enhancing the quality of PMSE observations by combining MAARSY and the Kilpisjärvi Atmospheric Imaging Receiver Array (KAIRA) in a multistatic approach. This observation type

would also enable the determination of a highly spatial-temporarily resolved windfield of the polar MLT.

## 5   Conclusions

The main conclusion from this study emphasizes that the extreme event reported by Chau et al. (2021) is certainly not unique. After analyzing a radar dataset obtained from MAARSY during the PMSE seasons between 2015 and 2021, this study successfully confirms the occurrence of additional high vertical velocity events. Over the course of 7 years, spanning 4 months

each, more than 700 incidents were identified that exhibited quasi-simultaneous upward and downward movements. Notably, a significant fraction of these events displayed absolute vertical velocities surpassing a threshold of 15 m/s ($3\sigma$), qualifying them as "extreme."

Although the majority of the recorded cases featured velocities below the aforementioned threshold, detailed statistical analyses were performed separately for varicose-mode cases in general and those with large vertical velocities. Our analyses

examined the occurrence statistics and vertical extent and vertical velocity characteristics of these events as well as their duration. It can be summarized that varicose-mode events occurred in every year covered by the study, regardless of the PMSE

month or time of day, with some variability in both aspects. The occurrence of varicose-mode events seems to align with the occurrence of PMSE, which in general held true for high-velocity events as well.

Further investigation into the duration, maximum vertical extent, and changes in vertical extent caused by the varicose structure ($h_1/h_0$) revealed an average duration of approximately 20 minutes (22 minutes for events with velocities exceeding 15 m/s), an extent of around 8 km (6.5 km for events with velocities exceeding 15 m/s), and a change in the maximum vertical extent of 50% (70% for events with velocities exceeding 15 m/s).

With 2.5%, the months June and July exhibit the heightest likelihood of encountering extreme events given the observation of PMSE, thereby offering a favorable opportunity to conduct in-depth investigations and gather valuable data related to such phenomena. In order to comprehensively capture and examine extreme events through a well-coordinated campaign, incorporating a multi-instrumental approach, the study at hand suggests that the optimal time window would fall within the months of June and July.

*Data availability.* The data to produce the figures are available in HDF5-format at https://www.radar-service.eu/radar/en/dataset/NNBGIyOIqWXJDgUG?token=oQYODhoohoMMgDJWfEff. The data to reproduce the figures 1 and 3 were taken from https://doi.org/10.1029/2021GL094918 (Chau et al., 2021).

*Author contributions.* JH identified, characterized, and analyzed the varicose-mode events described in the paper, and led the writing. JLC contributed the main idea for analyzing varicose-mode events and helped structure the paper's organization. RL conducted radar experiments with MAARSY and performed preliminary analyses. TRz aided in interpreting the study's findings. MZ assisted in reducing the extensive amount of complex radar voltages to specific channels and altitudes and provided spectral moment data from select years. All authors help improve early written versions of the paper.

*Competing interests.* The authors declare that they have no conflict of interest.

*Acknowledgements.* JH thanks Kesava Ramachandran, Dr. Fabio Feraco, and Dr. Miguel Urco for helpful discussions.



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
