# Peer review of "Observations of Polar Mesospheric Summer Echoes Resembling Kilometer-Scale Varicose-Mode Flows"

_EGUsphere, 2023_

## Author Comment (AC1)

**Response to RC1: 'Comment on egusphere-2023-1856', Anonymous Referee #1, 25 Sept. 2023**

**Major**

*I suggest to add more detailed definitions and method descriptions. One example is the vertical extent (l. 307). Is this the distance from the lowest to the highest altitude with significant PMSE detection? Is it the same as h1 (Fig. 7)? I also didn't find the resolution in altitude and time, i.e. the size of the "PMSE-pixels". This is an important information for someone who tries to replicate the work. As the patterns were detected manually, it is crucial to describe the process as detailed as possible for someone to repeat it. How exactly were meteor occurrences and system-induced artifacts discriminated from "good" data (l. 119)? How often did these artifacts occur? Please give more details on how the manual search was carried out (e.g. 20 min periods, +- 15 m/s, look for a minimum distance between h0 and h1, or for simultaneous up- and downdrafts of x m/s amplitude?). Some of it was described around l. 344, but it should be put into a dedicated paragraph in the Method section.*

➔ Thank you for your feedback! In response to your suggestions, we will include a dedicated paragraph in Chapter 2 to clarify the maximum vertical extent and its relationship to h1. We will also provide information on the temporal and spatial resolution of our measurements. The process for manually identifying patterns and discriminating meteor occurrences and system-induced artifacts from 'good' data will be described in more detail.

*Were there up- and downdrafts that were not simultaneous but slightly shifted?*

➔ Yes, indeed. This is precisely why we use the term "quasi-simultaneous".

*It would also be interesting to see a compilation of all or the most extreme events, e.g. those with a widening of 7 or beta of 6. Maybe this can be shown in a supporting information, or a zip file. It could be helpful for future research.*

➔ We will consider including this data in a supporting information file for future reference.

*The discussion leaves the impression that the work is unfinished, as it is a bit vague and unfocused (e.g. "the significance is yet to be determined", or the vague comparison with Hozumi et al., 2019). Suggestions for analysis that would be required to explain the origin of the structures are made but not carried out. How many events have simultaneous lidar data, and does that indicate temperature inversion layers? (For further investigations, it could be helpful to provide a list of the dates and times of the 707 events). Is there data on horizontal wind?*

➔ Thank you for your input! In the discussion, suggestions are given for possible approaches to the investigation of the physical causes, which are, however, not provided for in the scope of the present work. So far, we do not know for which observed events, measurements of other instruments like e.g. lidars or satellites are available. Despite the peak frequency of lidar measurements coinciding with our observation period, which transpires during the summer months, the ALOMAR lidar system is constrained by daylight conditions, limiting its altitude coverage to a maximum of 80 km. Other instruments as airglow imagers are contingent on nighttime conditions, which, regrettably, do not coincide with the observation of PMSE. Data on horizontal wind can and will be used in a follow-up paper including case studies for which we will look into those background conditions.

*It says "kilometer-scale" in the title, but no horizontal dimensions of the structures are estimated in the text.*

➔ Your finding is certainly correct. While the text does not provide specific measurements of the horizontal dimensions of the structures, the motivation for this study stemmed from the work of Chau et al., which did present radar imaging plots that demonstrated the spatial dimensions of similar events. The radar beam width imposes limitations on the observed area, and the presence of both upward and downward movements followed by 'no movement' (green) suggests that the areas traversing the beam with high vertical velocities cannot exceed a certain size. If the entire atmosphere within the observation volume were in motion, the patterns in the velocity plots would indeed look quite different. This provides context for the 'kilometer-scale' description in the title

*Inter- and intraannual variation of the occurrences of the structures are shown but remain fully unexplained.*

➔ It is our considered view that the current sample of events under examination may be insufficient in size to derive definitive conclusions. We must acknowledge that our observations are heavily contingent upon the presence of Polar Mesospheric Summer Echoes (PMSE), and thus, we exercise caution in formulating any conclusions. It is quite conceivable that high velocity events in the mesosphere, as visualized by our PMSE observations, occur throughout the year and show significant diurnal, seasonal or annual variability compared to our current findings.

*The diurnal variation of the occurrences of the structures is shown together with the diurnal variation of PMSE (Fig. 6), but the relative occurrence would be more interesting. Is there, or isn't there a significant diurnal or semidiurnal variation left when accounting for PMSE occurrence? Sect. 4.3 should be rewritten based on relative occurrences. Attention should be paid to units or resolution when stating occurrence rates. It is better to say "structures were observed on 33 out of 100 days on average" instead of "33%" (l. 245), because when counting occurrence or non-occurrence not on scales of days but on scales of 20 min or less, the occurrence rate is much smaller than 33%.*

➔ We appreciate your comment. We will explore the possibility of showing diurnal variations in relative occurrence and will revise and add to Section 4.3 accordingly. Your emphasis on units and resolution of occurrence rates is noted, and we will ensure that our reporting is more accurate.

**Comments by line number**

*l. 11 please add the total duration of the events, is it about 200 hours?*

➔ We can definitely add this value, thank you.

*l. 17 "highlighting their extreme nature": if so, you could mention that the distribution is non-Gaussian*

➔ Thank you. The fact is included as follows: Notably, a careful examination of the vertical velocities associated with these events confirmed that approximately 17% surpassed the 3σ threshold, highlighting their non-Gaussian distribution and extreme nature.

*Fig. 4 The examples show different numbers of oscillations, e.g. three in Fig. 4a. I would thus label t0, t1, t2,… (and not t0, t0, t0) to include a counter for this number in addition to your "d = duration" which is actually the period of one oscillation.*

➔ Thank you for your input, but we would prefer to maintain the current labeling. The intention behind the sketches is to illustrate how each individual event was identified. Even within a 'pair' or 'group,' each event is counted as a single occurrence.

*In Fig. 4c, only a slight variation can be seen in the evolution of the upper boundary, but the velocity measurements show a clear oscillation for several periods. The amplitude of these oscillations could also be an interesting parameter (it can be seen in the color, but a color bar is missing).*

➔ You are right, and indeed, the amplitude of these oscillations could be an interesting parameter to consider. However, implementing this change would extend beyond the scope of this current work..

*l. 120 Modelling is mentioned briefly, but it doesn't become clear what exactly was expected in terms of occurrence rate from the modeling.*

➔ Thank you for bringing this to our attention. In Feraco et al., it was found that 1 in 1000 events included vertical velocities outside the Gaussian reference. We will provide additional information on the expected occurrence rates in the upcoming sections. .

*l. 185 in addition to the mean and standard deviation, the kurtosis and skewness could be stated, that indicate in what way the distribution differs from a Gaussian.*

➔ We appreciate your suggestion. We will certainly explore the inclusion of kurtosis and skewness to better elucidate how the distribution differs from a Gaussian one. Thank you for this valuable input.

*Fig. 6b please calculate and give the percentages of the one-, two- and three-oscillation classes in the text, e.g. 80 %, 15 % and 5 %, and the same for the high-velocity subset.*

➔ Thank you for your suggestion. We will consider calculating and providing the percentages.

*Fig. 8a A logarithmic y axis might show better the extreme events with low counts.*

➔ Thank you for the recommendation. While a logarithmic y-axis might enhance the visibility of extreme events with low counts, it's worth noting that it could potentially create a misleading impression that these extreme values occur more frequently than they actually do.

*Fig. 8a Please add the Gaussian to Fig. 8a. Then one can directly see where it differs. I expect to see long tails, i.e. the extreme values are much more frequent than if the distribution would follow a Gaussian.*

➔ Thank you, we appreciate your input. However, it appears there may be a misunderstanding. In Fig. 8a, the distribution is intended to follow a Gaussian pattern, rather than differ from it. We will ensure this aspect is clarified in the text.

*In Fig. 8b you could show the Gaussian with the same vertical axis as on the left (so not scale it to fit in the window). So the peak will be way outside the plot, but then you can compare the tails, which is the interesting part. I think it is fine to show 8a with log y axis and 8b with linear y axis from 0 to 60 counts.*

➜ Thank you for your suggestions. We would prefer to maintain the current presentation for the same reasons as previously mentioned..

*Fig. 8b if the figure shows the histogram of the "maximum vertical velocities of varicose-mode events", then the total number should add up to 707. There are however many more. Is this the maximum w per profile? If so, what is the temporal resolution of a profile?*

➜ Thank you for your inquiry. Each 'wing' in the histogram does indeed add up to 707, as it should. This histogram represents two values for each event: the maximum updraft velocity and the maximum downdraft velocity. We will make sure to clarify this in the text. The temporal resolution of a profile will also be included for better context.

**Minor**

Your suggestions and the responses have been considered. If you have any more comments or questions, please feel free to ask.

*l. 105 the last part of the sentence is missing*

➜ Thank you! We missed that. We completed the sentence as follows …, which already modifies the temporal resolution and $f_N$ of the measurements.

*Fig. 4 Is it intentional that the colors appear somewhat unsharp? If not, my hint is that it is related to a problem with resolution when converting to or from postscript.*

➜ Thank you for your observation and suggestion regarding Fig. 4. The appearance of somewhat 'blurred' colors is, in fact, intentional. It's utilized to convey the signal strength, with brighter colors indicating stronger signals. This effect is particularly prominent in areas where the signal-to-noise ratio is lower.

*Fig. 4 please add a colorbar*

➜ Thank you for your suggestion to add a colorbar to Fig. 4. We will accommodate this request, although it's worth noting that the primary purpose of this figure is to illustrate the structural aspects rather than the velocities. Nevertheless, we will include a colorbar for additional clarity.

---

## Author Response (AR1)

**Final response to RC1: 'Comment on egusphere-2023-1856', Anonymous Referee #1, 25 Sept. 2023**

**We thank the reviewer for the thoughtful and valuable comments and questions. With the changes in the revised manuscript, we aim for more clarity.**

**Line references used in the comments by PC1 refer to the first version of the manuscript, while the line references used in the answers refer to the revised version of the manuscript. The comments by RC1 are colored grey and responses black. Actual changes in text are written in *italic*.**

I suggest to add more detailed definitions and method descriptions. One example is the vertical extent (l. 307). Is this the distance from the lowest to the highest altitude with significant PMSE detection? Is it the same as h1 (Fig. 7)? I also didn't find the resolution in altitude and time, i.e. the size of the "PMSE-pixels". This is an important information for someone who tries to replicate the work. As the patterns were detected manually, it is crucial to describe the process as detailed as possible for someone to repeat it. How exactly were meteor occurrences and system-induced artifacts discriminated from "good" data (l. 119)? How often did these artifacts occur? Please give more details on how the manual search was carried out (e.g. 20 min periods, +- 15 m/s, look for a minimum distance between h0 and h1, or for simultaneous up- and downdrafts of x m/s amplitude?). Some of it was described around l. 344, but it should be put into a dedicated paragraph in the Method section.

The resolution in time and altitude was added to line 123:

*"Within the observed period the altitude resolution is 300 m and the temporal resolution is on average $\approx$100 s depending on the experiment."*

To shed more light on the process of manually selecting varicose-mode events from the data set, including the determination of the vertical extent before and during each event ($h_1$ and $h_2$) and the process of deciding on "good" and "bad" data, we added the following paragraph after from line 140:

*"With these denotations, it is attempted to follow those in Figure 1a (yellow vertical lines). The selection of varicose-mode events was a manual procedure mainly using the daily vertical velocity plots with a color bar chosen to range from -15 to 15 m/s to focus on events exceeding the 3$\sigma$ threshold. The data analysis involved an initial step of data masking, where only data points meeting specific criteria were retained for visualization. These criteria included a signal-to-noise ratio (SNR) not falling below -5 and vertical velocities not exceeding 100 m/s. Consequently, data points falling outside of these established limits were excluded from the representation. This selective approach allowed for the clear depiction of PMSE. By zooming in on the figure, the temporal localization of events became more apparent, enabling the identification of event boundaries and the determination of the upper and lower extents of PMSE before, during, and after each event. For instance, the PMSE layer's width immediately prior to an event's initiation ($t_0$) was denoted as $2h_0$ while $2h_1$ represented the maximum width of the event, measured from the first pixels at the upper and lower edges. Typically, the widest point corresponds to the juncture of the initial updraft and downdraft phases within the event. In some instances, updrafts and downdrafts did not occur simultaneously but rather quasi-simultaneously, exhibiting a slight temporal offset relative to each other. In such cases, the midpoint of the event might not coincide with the point of maximum width. Upon identifying and characterizing an event within its spatial and temporal dimensions, further steps were necessary to determine the maximum vertical velocities. As previously mentioned, employing an automated search method posed challenges due to the sporadic occurrence of meteors. These meteors have the potential to not only amplify the signal but also yield exceptionally high vertical velocities. To ensure that these meteor events did not influence the study of*

*varicose-mode events, the power-time plots forming the basis of the spectra used for velocity determination were meticulously examined for characteristics primarily associated with short-term, intense signals across multiple altitude channels. When a presumed velocity maximum coincided with such a signature in the power-time plot, it was disregarded, and the subsequent smaller value was considered."*

Were there up- and downdrafts that were not simultaneous but slightly shifted?

Yes, indeed. This is precisely why we use the term "quasi-simultaneous".

It would also be interesting to see a compilation of all or the most extreme events, e.g. those with a widening of 7 or beta of 6. Maybe this can be shown in a supporting information, or a zip file. It could be helpful for future research.

We will consider including this data in a supporting information file for future reference.

The discussion leaves the impression that the work is unfinished, as it is a bit vague and unfocused (e.g. "the significance is yet to be determined", or the vague comparison with Hozumi et al., 2019). Suggestions for analysis that would be required to explain the origin of the structures are made but not carried out. How many events have simultaneous lidar data, and does that indicate temperature inversion layers? (For further investigations, it could be helpful to provide a list of the dates and times of the 707 events). Is there data on horizontal wind?

Thank you for your input! In the discussion, suggestions are given for possible approaches to the investigation of the physical causes, which are, however, not provided for in the scope of the present work. So far, we do not know for which observed events, measurements of other instruments like e.g. lidars or satellites are available. To the manuscript we added from line 280:

*"Despite the peak frequency of lidar measurements coinciding with our observation period, which transpires during the summer months, the ALOMAR lidar system is constrained by daylight conditions, limiting its altitude coverage to a maximum of 80 km. The Iron resonance lidar from IAP was only operational until 2017. For these reasons, the number of measurements useful for a study of temperatures during the occurrence of PMSE is very scarce."*

Data on horizontal wind can and will be used in a follow-up paper including case studies for which we will look into those background conditions.

It says "kilometer-scale" in the title, but no horizontal dimensions of the structures are estimated in the text.

Your finding is certainly correct. While the text does not provide specific measurements of the horizontal dimensions of the structures, the motivation for this study stemmed from the work of Chau et al., which did present radar imaging plots that demonstrated the spatial dimensions of similar events. The radar beam width imposes limitations on the observed area, and the presence of both upward and downward movements followed by 'no movement' (green) suggests that the areas traversing the beam with high vertical velocities cannot exceed a certain size. If the entire atmosphere within the observation volume were in motion, the patterns in the velocity plots would indeed look quite different. This provides context for the 'kilometer-scale' description in the title.

Inter- and intraannual variation of the occurrences of the structures are shown but remain fully unexplained.

It is our considered view that the current sample of events under examination may be insufficient in size to derive definitive conclusions. We must acknowledge that our observations are heavily

contingent upon the presence of Polar Mesospheric Summer Echoes (PMSE), and thus, we exercise caution in formulating any conclusions. It is quite conceivable that high velocity events in the mesosphere, as visualized by our PMSE observations, occur throughout the year and show significant diurnal, seasonal or annual variability compared to our current findings.

The diurnal variation of the occurrences of the structures is shown together with the diurnal variation of PMSE (Fig. 6), but the relative occurrence would be more interesting. Is there, or isn't there a significant diurnal or semidiurnal variation left when accounting for PMSE occurrence? Sect. 4.3 should be rewritten based on relative occurrences. Attention should be paid to units or resolution when stating occurrence rates.

In the revised version of the manuscript, we changed Figure 6a. It now shows the diurnal variability of varicose mode events with respect to the PMSE occurrence. Since no changes in distribution are visible towards the version before (minima, maxima), additional changes to the text were not necessary. The only change made is to replace *"absolute"* with *"relative"* in line 202.

It is better to say "structures were observed on 33 out of 100 days on average" instead of "33%" (l. 245), because when counting occurrence or non-occurrence not on scales of days but on scales of 20 min or less, the occurrence rate is much smaller than 33%.

This comment was addressed as follows (line 293):

*"The analysis of more than 700 varicose structures in the MLT region revealed an overall occurrence of approximately 3 in 100."*

l. 11 please add the total duration of the events, is it about 200 hours?

This comment is addressed by adjusting the sentence in line 12 as follows:

*"Over the seven-year period, we observed and recorded more than 700 varicose-mode events with a total duration of about 265 hours."*

l. 17 "highlighting their extreme nature": if so, you could mention that the distribution is non-Gaussian

Thank you. The fact is included as follows (line 17):

*"Notably, a careful examination of the vertical velocities associated with these events confirmed that approximately 17% surpassed the 3σ threshold, highlighting their non-Gaussian distribution and extreme nature."*

Fig. 4 The examples show different numbers of oscillations, e.g. three in Fig. 4a. I would thus label t0, t1, t2,… (and not t0, t0, t0) to include a counter for this number in addition to your "d = duration" which is actually the period of one oscillation.

Thank you for your input, but we would prefer to maintain the current labeling. The intention behind the sketches is to illustrate how each individual event was identified. Even within a 'pair' or 'group,' each event is counted as a single occurrence.

In Fig. 4c, only a slight variation can be seen in the evolution of the upper boundary, but the velocity measurements show a clear oscillation for several periods. The amplitude of these oscillations could also be an interesting parameter (it can be seen in the color, but a color bar is missing).

You are right, and indeed, the amplitude of these oscillations could be an interesting parameter to consider. However, implementing this change would extend beyond the scope of this current work.

l. 120 Modelling is mentioned briefly, but it doesn't become clear what exactly was expected in terms of occurrence rate from the modeling.

Thank you for bringing this to our attention. We decided to not mention occurrence rates of the extreme events due to considerable difference between the background conditions used in the simulations and those of the MLT region. For instance: even though the Froude number used in the NS performed by Feraco et al. (2018) was realistic for the MLT, in each run it was kept constant in the entire spatial domain. On the contrary, in the atmosphere, the Froude number evolves with time and space. For this reason, it is not reasonable to extract an occurrence rate made by Feraco et al. (2018).

l. 185 in addition to the mean and standard deviation, the kurtosis and skewness could be stated, that indicate in what way the distribution differs from a Gaussian.

Thank you very much for this valuable input. We gave it a lot of thought and to make everything as transparent as possible we made the following changes:

Since tea vast majority of the vertical velocity values are 0 in our measurements we "cut off" the peak of the distribution in Fig. 8a) in order to fit a reasonable Gaussian distribution. The kurtosis and skewness does not describe the distribution reasonably, which is why we decided to not include those values in the manuscript.

Fig. 6b please calculate and give the percentages of the one-, two- and three-oscillation classes in the text, e.g. 80 %, 15 % and 5 %, and the same for the high-velocity subset.

This comment is addressed by adding the percentages for the general varicose-mode events as well as for the high-velocity events. The changes can be found from line 204:

*"In case of Figure 6b the number of events with |w| ≥ 15 m/s is increased by one count if a single event within a pair or group has such high vertical velocities. In the varicose mode, single occurrences (371 events, 52% of the total) were observed predominantly, followed by event pairs (92 pairs, 26% of the total). The least frequent are groups, with 46 counted instances (22% of the total), which contain 3 to 8 events within a brief time frame."*

And from line 242:

*"Specifically, 65 events were counted as single occurrences, constituting approximately 9% of the total count of single events. For event pairs, the proportion is approximately 20%, while group instances account for around 47%. Transitioning from single events to event groups, there is a decrease in the absolute number of occurrences within each category, while the proportion of high-velocity events increases."*

Fig. 8a A logarithmic y axis might show better the extreme events with low counts.

Thank you for the recommendation. While a logarithmic y-axis might enhance the visibility of extreme events with low counts, it's worth noting that it could potentially create a misleading impression that these extreme values occur more frequently than they actually do.

Fig. 8a Please add the Gaussian to Fig. 8a. Then one can directly see where it differs. I expect to see long tails, i.e. the extreme values are much more frequent than if the distribution would follow a Gaussian.

To clarify this issue, we changed Figure 8. And it's caption as follows:

*"**Figure 8:** Distribution (a) of vertical velocities across all altitudes and throughout the observation period and (b) of the maximum vertical velocities of varicose-mode events in blue (down-drafts) and red (up-drafts). In both panels, the black curve represents the Gaussian reference."*

And we made the respective change in the text in line 230:

*"The black line compares these values to the Gaussian reference, mirroring the distribution in Figure 8a."*

In Fig. 8b you could show the Gaussian with the same vertical axis as on the left (so not scale it to fit in the window). So the peak will be way outside the plot, but then you can compare the tails, which is the interesting part. I think it is fine to show 8a with log y axis and 8b with linear y axis from 0 to 60 counts.

Thank you for your suggestions. We would prefer to maintain the current presentation for the same reasons as previously mentioned.

Fig. 8b if the figure shows the histogram of the "maximum vertical velocities of varicose-mode events", then the total number should add up to 707. There are however many more. Is this the maximum w per profile? If so, what is the temporal resolution of a profile?

Thank you for your inquiry. Each 'wing' in the histogram does indeed add up to 707, as it should. This histogram represents two values for each event: the maximum updraft velocity and the maximum downdraft velocity. We will make sure to clarify this in the text. The temporal resolution of a profile will also be included for better context.

To clarify, we adjusted line 218 to:

*"Figure 8b shows the velocity distributions for the highest upward and downward velocities recorded during each varicose-mode event (two values per event), represented by red and blue bars, respectively."*

l. 29 remove ", e.g. the so-called"

change (line: 31): *"Ice particles larger than ≈ 20 nm can be observed with the naked eye as, e.g. noctilucent clouds (NLC), at an altitude of about 82 km."*

l. 30 change "high" to "mid" (latitudes). NLC occur at high latitudes, but can only be visually observed in twilight (not at night!) and thus from a mid latitude viewing towards north. At high latitudes the sky is too bright for naked-eye observers to see them.

Change (line 32): *"These clouds are visible to the observer at high- and mid-latitudes as they are illuminated by the sun that has set during twilight and the night."*

l. 66 the reference to Taylor et al., 1995 should be added here as well

Reference added

l. 72 sentence with "tradionally" and "outliers" is double

corrected

l. 76 delete "even"

deleted

l. 105 the last part of the sentence is missing

Thank you! We missed that. We completed the sentence as follows in line 113:

*"Exceptions from that are three months (June-August) in 2018, one month (July) in 2019, and one month (July) in 2020 where the PRF was adjusted to 828 Hz, 850 Hz and 900 Hz, respectively, which modifies the temporal resolution and $f_N$ of the measurements."*

Fig. 1 remove "of the highlights" in the first sentence of the caption

removed

Fig. 3: The labels a) and b) are missing in the figure.

added

Fig. 3 Is it Fig. 3a or b that the velocities are missing? Please check that the text is correct.

Thanks, we corrected that.

Fig. 3 Please add the date of this observation.

Of course, we added the date.

Fig. 4 Is it intentional that the colors appear somewhat unsharp? If not, my hint is that it is related to a problem with resolution when converting to or from postscript.

Thank you for your observation and suggestion regarding Fig. 4. The appearance of somewhat 'blurred' colors is, in fact, intentional. It's utilized to convey the signal strength, with brighter colors indicating stronger signals. This effect is particularly prominent in areas where the signal-to-noise ratio is lower.

Fig. 4 please add a colorbar

We added it.

Fig. 4 dot missing at end of caption

Dot added.

Fig. 4 caption second line: change downwards to downward

Changed

l. 165 "2.5% in June and July, 0.3% in May and 1.0% in August"

changed (see line 201)

l. 193 remove "in" after "observed"

changed

Fig. 7 Horizontal axis numbers are easier to read

Changed to horizontal numbers

Fig. 8 Labels a and b are missing in figure

Thank you! We overlooked that and added the labels now.

l. 195 in a number of places, you explain about the more intense colors and thicker outlines of the bars for the values above 15 m/s. I would maybe mention this in the first caption where it applies (Fig. 5 I think) stating that it is done like this in the following figures also, and maybe once in the text, but not over and over again.

We followed you advice. The explanation of the thicker lines and stringer colors is given once in line 189.

l. 200 delete"," after "found"

deleted

l. 200 delete "also"

deleted

l. 202 no brackets around the two values

changed

l. 211 occurr -> occur

corrected

l. 295 delete "relatively narrow (". Just give the numbers. Some readers might not consider 5.4 km to be narrow. Or, if you want to stress it, you should say compared to what the width is narrow, e.g. compared to other radars of that type if that is true and potentially important e.g. to find out if the structures can or cannot be found in other radar datasets.

Thank you for your input! We decided on keeping the sentience as is.

l. 398 "heightest" -> highest

corrected

**Final response to RC2: 'Comment on egusphere-2023-1856', Anonymous Referee #2, 03 Oct. 2023**

**We thank the reviewer for the thoughtful and valuable comments and questions. With the changes in the revised manuscript, we aim for more clarity.**

**Line references used in the comments by PC1 refer to the first version of the manuscript, while the line references used in the answers refer to the revised version of the manuscript. The comments by RC1 are colored in grey and responses in black. Actual changes in text are written in italic.**

Lines 63 and Section 4.1: It is difficult (for me) to see how the varicose-mode extreme events in Figures 1 and 4 visually resemble a mesospheric bore/ soliton. As mentioned in section 4.1 mesospheric bores are characterized by a sharp wave crest followed by smaller trailing waves. While there are some solitary waves structures for e.g. in Figure 4 in the overall vertical velocity evolution, the varicose events i.e. upwards/downwards velocity (e.g. Figure 4c at 10:10 UT) don't appear to have any resemblance to a mesospheric bore (sharp wave crest/solitary wave).

Thank you for your observation and comment. We understand your point of view, and we have taken your feedback into consideration. In Section 4.1, we have clarified that in the selected, specific cases, the bore-defining feature, which is the presence of trailing waves, is indeed missing. This absence of trailing waves distinguishes the varicose-mode extreme events from typical mesospheric bores or solitons.

Additionally while the text in Section 4.1 alludes to possible coincident observations to infer background conditions to investigate potential ducting mechanisms, no actual data is presented. I think it would be useful to the reader if the authors could clarify how the extreme varicose events are similar to a mesospheric bore. Perhaps provide more information and/or expand on Lighthill, (1979) [Line 230].

Thank you for your feedback. In response to your point, we plan to present background conditions, including actual data, for some cases in dedicated case studies in a follow-up paper. In the current version of the paper, the use of the theory of bores/solitons serves as an initial attempt to provide a foundation for exploring potential explanations for these previously unknown structures. As discussed in Section 4.1, while the resemblance in structure hints at potential underlying processes such as a guiding mechanism linked to a thermal or Doppler duct, we intend to delve deeper into these aspects in future research.

Line 39: 'Those' refers to PMSE?

Yes, indeed.

Figure 1. What do the vertical yellow lines in Figure 1a indicate? Please mention what 'SW' refers to.

The yellow vertical lines illustrate the vertical width of the PMSE layer before and during the extreme event, later denoted as h1 and h2. We added this information to the paragraph of the text in line 140:

*"With these denotations, it is attempted to follow those in Figure 1a (yellow vertical lines)."*

Lines 72-74. Repetitive lines. Maybe delete one of the two lines. Maybe I missed it, but Chau et al., (2021) mention five standard deviations and not why they use three standard deviations to identify outliers. Please provide additional references where '3 sigma' has been traditionally used.

Repetition is corrected (see line 81).

You are absolutely right, we changed the reference to (see line 82):

*Lehmann, R.: 3σ-Rule for Outlier Detection from the Viewpoint of Geodetic Adjustment, Journal of Surveying Engineering, 139, 157–165, https://doi.org/10.1061/(ASCE)SU.1943-5428.0000112, 2013.*

Line 105: Incomplete sentence

Completed the sentence as follows (see line 113):

"*Exceptions from that are three months (June-August) in 2018, one month (July) in 2019, and one month (July) in 2020 where the PRF was adjusted to 828 Hz, 850 Hz and 900 Hz, respectively, which modifies the temporal resolution and $f_N$ of the measurements.*"

Figure 3 caption- last line: Should this be 'missing in (a)' ?

You are right, thank you! We corrected that in the figure caption.

Figure 4: I am curious as to why in Figure 4c, right column, the decreasing layer thickness after the increase (after blue color at the bottom), is not indicative of upward velocity?

Thank you for your question. The reason for the decreasing layer thickness after the increase (following the blue color at the bottom) in Figure 4c is that, in the data, there was no updraft measured during this time period.

Line 185: From Figure 8a, even with closer inspection it is difficult to observe "that downward vertical velocities (negative values) predominate". Would a zero line and/or some numerical values help reveal this conclusion? Or is this referring to discussion in Sec. 3.4?

This observation can only be made by looking into the results in very detail. Even a black line pointing out 0 would not make a difference in this case. This observation was made in the data analysis process and we found it interesting that we were able to see this well known effect in our observations as well.

Line 193: 'were observed in simultaneously in' --> were observed simultaneously in

Thank you, we corrected that.

Line 211: "do not occurr frequent"--> occur frequently

Thank you! We corrected it.

---

## Author Response (AR2)

**Response to Anonymous Referee #1**

**Minor**

The authors state that the altitude resolution is 300~m yet Fig. 4c shows a double layer structure with 200 m distance and 80 m width. Maybe the vertical grid is higher and the profiles were smoothed with a 300~m window?

Thank you for bringing that to my attention! Indeed, while the resolution is 300 meters, the color brightness in the plots depicted in Figure 4 is dependent upon the signal-to-noise ratio of the received PMSE. In selecting the color bar for the plots, we aimed to show the up- and down-drafts prominently while preventing color oversaturation. Furthermore, the values undergo interpolation, contributing to the perception of a higher range resolution. However, when plotting solely the velocities or the signal-to-noise ratio from the uploaded plot data, the actual resolution becomes more evident.

In l. 148 "(yellow vertical lines)" is missing.

Thank you! We added it.

In contrast to the response document, no paragraph was added from line 280.

Thank you very much for the comment! It was forgotten to remove this answer from the final response, we are sorry. We decided to leave this paragraph out completely.

I do not fully understand the discussion following my remark that it says "kilometer-scale" in the title but that no horizontal dimensions are estimated in the text. Is the argument that the horizontal wind is zero on average? Is that realistic? I still feel some deduction of horizontal scale should be included in the text when "kilometer-scale" is stated so prominently in the title.

The term "kilometer-scale" does not correspond to the fact that the velocities are so small. We are using this wording because earlier research showed these structures are kilometer-sized (Chau et al. 2021). In our study, we are using the term "Kilometer-scale" to connect our findings and our paper to what was found before.

To make it more clear to the reader we added additional information in line 64: *"Figure panel c illustrates a two-dimensional plane from radar imaging, offering spatial information on the event. The up- and downdrafts exhibit clear localization both horizontally and vertically, measuring 3–4 km in width along the x-axis and extending at least 8–12 km along the y-axis."*

and in line 86: *"(referring to the spatial information taken from the radar imaging of the extreme event by Chau et al. 2021).*